# Hydration Status, Fluid Intake, Sweat Rate, and Sweat Sodium Concentration in Recreational Tropical Native Runners

**DOI:** 10.3390/nu13041374

**Published:** 2021-04-20

**Authors:** Juthamard Surapongchai, Vitoon Saengsirisuwan, Ian Rollo, Rebecca K. Randell, Kanpiraya Nithitsuttibuta, Patarawadee Sainiyom, Clarence Hong Wei Leow, Jason Kai Wei Lee

**Affiliations:** 1Faculty of Physical Therapy, Mahidol University, Nakhon Pathom 73170, Thailand; 2Department of Physiology, Faculty of Science, Mahidol University, Bangkok 10400, Thailand; vitoon.sae@mahidol.ac.th (V.S.); kanpiraya.sutt@gmail.com (K.N.); psonepp@gmail.com (P.S.); 3Gatorade Sports Science Institute, PepsiCo Life Sciences, Global R&D, Leicestershire LE4 1ET, UK; ian.rollo@pepsico.com (I.R.); rebecca.randell@pepsico.com (R.K.R.); 4School of Sport, Exercise and Health Sciences, Loughborough University, Leicestershire LE11 3TU, UK; 5Human Potential Translational Research Programme, Yong Loo Lin School of Medicine, National University of Singapore, Singapore 119283, Singapore; medhwcl@nus.edu.sg (C.H.W.L.); phsjlkw@nus.edu.sg (J.K.W.L.); 6Department of Physiology, Yong Loo Lin School of Medicine, National University of Singapore, Singapore 117593, Singapore; 7N.1 Institute for Health, National University of Singapore, Singapore 117456, Singapore; 8Global Asia Institute, National University of Singapore, Singapore 119076, Singapore; 9Institute for Digital Medicine, Yong Loo Lin School of Medicine, National University of Singapore, Singapore 117456, Singapore; 10Singapore Institute for Clinical Sciences, Agency for Science, Technology and Research (A*STAR), Singapore 117609, Singapore

**Keywords:** recreational running, tropical climate, sweat electrolyte, fluid replacement, hydration plan

## Abstract

Aim: The purpose of this study was to evaluate hydration status, fluid intake, sweat rate, and sweat sodium concentration in recreational tropical native runners. Methods: A total of 102 males and 64 females participated in this study. Participants ran at their self-selected pace for 30–100 min. Age, environmental conditions, running profiles, sweat rates, and sweat sodium data were recorded. Differences in age, running duration, distance and pace, and physiological changes between sexes were analysed. A *p*-value cut-off of 0.05 depicted statistical significance. Results: Males had lower relative fluid intake (6 ± 6 vs. 8 ± 7 mL·kg^−1^·h^−1^, *p* < 0.05) and greater relative fluid balance deficit (−13 ± 8 mL·kg^−1^·h^−1^ vs. −8 ± 7 mL·kg^−1^·h^−1^, *p* < 0.05) than females. Males had higher whole-body sweat rates (1.3 ± 0.5 L·h^−1^ vs. 0.9 ± 0.3 L·h^−1^, *p* < 0.05) than females. Mean rates of sweat sodium loss (54 ± 27 vs. 39 ± 22 mmol·h^−1^) were higher in males than females (*p* < 0.05). Conclusions: The sweat profile and composition in tropical native runners are similar to reported values in the literature. The current fluid replacement guidelines pertaining to volume and electrolyte replacement are applicable to tropical native runners.

## 1. Introduction

Sweat evaporation is important for the dissipation of metabolic heat production, which may increase ten- to twenty-fold during exercise [1]. In hot environments, evaporative sweat cooling is the main avenue of heat loss, preventing rapid rises of core body temperature [2,3,4,5]. Hypohydration, experienced as a result of sweat loss, increases physiological strain and perception of effort, which can decrease endurance exercise performance [6,7]. In addition, sweat loss during exercise can also result in electrolyte imbalance such as hyponatremia. Thus, it is important to replace electrolyte losses as part of the rehydration process after exercise [3,8,9].

Running is a common form of exercise as it can be easily performed and does not require any specialised equipment. There are an estimated five to eight million individuals participating in running events globally. A 50% increase in participation in running has been tracked over the last decade. This growth has, in part, been driven by increased participation in Asia [10]. In tropical warm, high humidity environments, the evaporation of sweat may be compromised, leading to lower rates of body heat dissipation [11]. Thus, it is reasonable to suggest that appropriate hydration may be even more important for endurance running when in tropical Asian countries.

An athlete’s sweat rate and sweat electrolyte concentration vary depending on individual characteristics, type and intensity of exercise, clothing, equipment worn as well as environmental conditions [12,13,14,15,16,17,18,19,20]. Therefore, the assessment of individual sweat rate and sweat electrolytes losses for specific exercise and environmental conditions is recommended to create individualised hydration strategies. This approach may reduce the risk of heat illness and optimise performance [21]. Despite the individuality in the response to dehydration, current guidelines advise limiting fluid deficits to no more than 2% body mass loss during exercise to avoid compromised cognitive function and aerobic exercise performances [21]. Further decrements in performance are associated with increasing levels of hypohydration (3–10%), particularly in hot weather typical in tropical climates [12,22]. Individualised drinking plans may also reduce the risk of over-drinking and exercise-associated hyponatremia [23,24,25,26,27,28].

Tropical natives are likely to be more heat-acclimatised than athletes who live in temperate or cool environments. Heat-acclimatised individuals have thermoregulatory adaptations such as lowered core body temperature, lowered heart rate, earlier onset of sweating and higher sweat rate [29,30,31]. Although variable, mean sweat sodium concentrations ([Na^+^]) are reported to be approximately 50 mmol·L^−1^ [32]. To the authors’ knowledge, there are limited data on sweat rate and sweat composition of tropical native athletes, which may impact on hydration strategies during and after exercise in this population. Knowledge of sweat responses and sweat composition of tropical native athletes will allow us to understand if consensus recommendations on hydration are also relevant to heat-acclimatised athletes.

Therefore, the purpose of the present study was to evaluate the hydration status, fluid intake, sweat rate, and sweat [Na^+^] in recreational tropical native runners.

## 2. Materials and Methods

### 2.1. Study Design and Participants

This study adopted an observational cohort study design. A total of 102 males and 64 females were recruited to participate in this study. All measurements were made on a single day of practice sessions in various running groups. Ethics approval was obtained from the Centre of Ethical Reinforcement for Human Research, Mahidol University (MU-CIRB 2018/208.2311 and protocol no. 2018/198.0910). All participants provided written informed consent before participation.

### 2.2. Experimental Protocol

Resting heart rate, blood pressure and aural temperature were measured before and after running. Heart rate and blood pressure were measured using an upper arm blood pressure monitor (BM 28, Beurer GmbH, Ulm, Germany). Aural temperature was measured using an ear thermometer (FT 78, Beurer GmbH, Ulm, Germany). Participants with high resting blood pressure and/or high resting aural temperature (systolic blood pressure >180 mmHg and/or diastolic blood pressure >110 mmHg and/or aural temperature >38 °C) were excluded from the study.

Six running sessions were completed on separate days. Each session involved a warm-up of 10–15 min, followed by 30–70 min of running, and ended with 10–15 min of cool-down. Participants ran at their individual pace, with most participants running at a light to moderate intensity. Two running sessions were conducted in the morning, between 6 a.m. and 8 a.m., while four running sessions were conducted in the evening, between 5 p.m. and 8 p.m. The first five running sessions were conducted at Lumphini Park, Bangkok, Thailand while the sixth running session was conducted in a park within Mahidol University, Nakhon Pathom, Thailand. All six sessions were held in a public park with a 2.5 km running track. A water station was provided. Runners consumed plain water (Aquafina, PepsiCo, Harrison, NY, USA) *ad libitum* from individual water bottles. Water bottles were weighed before and after each running session to record the volume of fluid intake during running.

Before each running session, all participants voided their bladders. Mid-stream urine samples were collected to measure urine specific gravity (USG). Pre-exercise body mass was measured using a bench scale (N.V. Mettler-Toledo S.A., Zaventem, Belgium) while minimally clothed (T-shirts, shorts or tights, and without shoes), and recorded to the nearest 0.10 kg. For participants who needed to urinate during the run, body mass was measured before and after the excretion to estimate urine output. To collect sweat, the right or left forearm was cleaned with an alcohol pad (3M, Minneapolis, MN, USA), rinsed with distilled water, and dried with electrolyte-free gauze. An absorbent patch (9 cm × 10 cm) (3M™ Tegaderm™ + Pad Film Dressing with Non-Adherent Pad, 3M, Minneapolis, MN, USA) was then applied to the mid-forearm [33].

After each running session, the participants towel-dried themselves and post-exercise body mass was measured. The same bench scale was used and participants wore the same attire as during the pre-exercise body mass assessment. The absorbent patch was then removed from the forearm, placed in the barrel of a plastic syringe using clean forceps and squeezed with a plunger to collect sweat. Sweat samples were analysed for sweat [Na^+^] and sweat potassium concentration ([K^+^]).

### 2.3. Measurement of Environmental Conditions

Ambient temperature and relative humidity were measured and recorded at 10-min intervals using a data logger (QUESTemp°34, 3M, Minneapolis, MN, USA) during each running session, and the mean value was calculated.

### 2.4. Urine Specific Gravity (USG)

USG from mid-stream urine samples were measured using a hand-held refractometer (PAL-10S, ATAGO^®^, Saitama, Japan). USG was assessed in duplicates and the average value was used for recording. USG was used an indicator of hydration status, with USG >1.020 indicating hypohydration and USG >1.030 indicating severe hypohydration [34].

### 2.5. Whole-Body Sweat Loss (WBSL) and Whole-Body Sweat Rate (WBSR)

WBSL and WBSR were calculated using Equations (1) and (2) respectively:WBSL (L) = (Pre-exercise body mass (kg) − Post-exercise body mass (kg)) + Fluid intake (L) − Urine output (L),(1)
WBSR (L·h^−1^) = WBSL (L)/Exercise duration (h).(2)

### 2.6. Whole-Body Sweat Sodium Concentration

Sweat [Na^+^] and sweat [K^+^] were analysed via ion-selective electrode (ISE) technology using Na^+^ (LAQUAtwin Na-11, HORIBA Advanced Techno Co., Ltd., Kyoto, Japan) and K^+^ analysers (LAQUAtwin K-11, HORIBA Advanced Techno Co., Ltd., Kyoto, Japan). Whole-body sweat [Na^+^] (mmol·L^−1^) and whole-body sweat Na^+^ loss (mmol) were calculated using Equations (3) and (4) respectively [33]:Predicted whole-body sweat [Na^+^] (mmol·L^−1^) = 0.57 (forearm sweat Na^+^) + 11.05,(3)
Whole-body sweat Na^+^ loss (mmol) = WBSL (L) ∗ Predicted whole-body sweat [Na^+^] (mmol·L^−1^).(4)

Sweat [Na^+^] were classified into three groups: low ([Na^+^] <30 mmol·L^−1^), moderate ([Na^+^] = 30–60 mmol·L^−1^), and high ([Na^+^] >60 mmol·L^−1^) [35,36]. Additionally, 12 samples were randomly selected and analysed using the gold standard high-performance liquid chromatography (HPLC) method (Dionex ICS-5000, Thermo Fisher Scientific, Inc., Waltham, MA, USA).

### 2.7. Statistical Analysis

Statistical analysis was conducted using IBM SPSS Statistics version 19.0 (IBM, Armonk, NY, USA). Descriptive data were generally expressed as mean ± standard deviation (SD). All biochemical data were log-transformed to reduce non-uniformity of error. The data were back transformed before being expressed as parametric mean ± SD. Normality was determined using the Shapiro–Wilk test. Differences in age, running characteristics, body mass loss, fluid intake, and net fluid balance between males and female runners were analysed using independent *t*-test. Pearson correlation coefficient was used to analyse the correlation between parameters, including the correlation of whole-body sweat [Na^+^] between ISE and HPLC methods. Correlation coefficients were interpreted based on the following thresholds: *r* ≤ 0.35 = weak, 0.36 ≤ *r* ≤ 0.67 = moderate, and 0.68 ≤ *r* ≤ 1.0 = strong [37]. For all analyses, a *p*-value of < 0.05 was considered significant.

## 3. Results

### 3.1. Number of Subjects and Environmental Conditions for Each Running Session

There were 166 participants in total (102 males and 64 females). The number of participants for each running session is shown in Table 1, together with the mean ambient temperature and mean relative humidity. The mean (range) ambient temperature and relative humidity across the six running sessions were 29.6 (28.0–31.5) °C and 70 (55–87)% respectively.

### 3.2. Participants’ Age, Running Profile, and Body Mass Change across Running

Participants were aged between 21–68 years with running experience ranging from 6 months to more than 10 years. All runners were native to Thailand and had been within the country for 6 months, exercising in hot and humid environments, prior to the trial. The running durations of all participants during the running sessions were between 30–100 min. Male runners ran a further mean distance and at a faster mean pace than female runners (*p* < 0.05) (Table 2). However, the mean running duration did not differ between sexes. Both mean WBSL and mean WBSR among the male runners were greater than among the female runners (*p* < 0.05) (Table 2). While six male and four female runners had >2% body mass loss after the running session, percentage body mass loss did not differ between sexes (*p* > 0.05) (Table 2).

### 3.3. Urine Specific Gravity

USG data were absent from 18 male and four female runners due to insufficient urine samples. Mean USG of runners who ran in the morning and evening were 1.015 ± 0.008 and 1.013 ± 0.007, respectively. There was no difference between the mean USG of runners from the morning and evening sessions (*p* > 0.05). The number of hypohydrated participants (USG > 1.020) did not differ between the morning or evening running session (*p* > 0.05). However, a greater percentage of participants were hypohydrated (USG > 1.020) before the run when the running session was conducted in the morning (28%) than in the evening (15%) (Table 3). A greater number of runners were severely hypohydrated (USG > 1.030) before the run when the session was conducted in the morning (4%) than in the evening (1%). In addition, when the running session was conducted in the morning, there is a moderate positive correlation between pre-exercise USG and fluid intake (*p* < 0.05, *r* = 0.42), and a moderate negative correlation between pre-exercise USG of male runners and WBSL (*p* < 0.05, *r* = −0.54).

### 3.4. Relative Sweat Loss and Fluid Intake during Running

Mean WBSR during recreational running was higher in males than females (*p* < 0.05) (Table 2). Sweat loss relative to body mass was calculated and differences between sexes were compared. Males had higher sweat loss relative to body mass than female runners (19 ± 8 vs. 16 ± 6 mL·kg^−1^·h^−1^, *p* < 0.05) (Figure 1). With regards to fluid intake during running, 15 of the 102 male runners (14.7%) did not drink any water during the run. However, only two of the 64 female runners (3.1%) did not drink any water during the run. Mean *ad libitum* fluid intake during running in males and females did not differ between sexes (*p* > 0.05) (Table 2). However, male runners had a lower fluid intake relative to body mass than female runners (6 ± 6 vs. 8 ± 7 mL·kg^−1^·h^−1^, *p* < 0.05) (Figure 1). Therefore, males had a greater negative fluid balance relative to body mass than female runners (−13 ± 8 mL·kg^−1^·h^−1^ or −67.4% vs. −8 ± 7 mL·kg^−1^·h^−1^ or −48.9%, *p* < 0.05) (Figure 1). Furthermore, running pace and sweat rate were found to have a moderate negative correlation (*r* = −0.47, *p* < 0.01).

### 3.5. Sweat Sodium and Potassium Concentration

Sweat data were absent from five male and 15 female participants due to insufficient volume of sweat sample. Figure 2A presents the distribution of sweat [Na^+^] among the remaining 97 males and 49 female runners. Sweat [Na^+^] did not correlate with sweat rate, USG, fluid intake, or running distance or pace (*p* > 0.05). A strong positive correlation was observed between sweat [Na^+^] measured using ISE method and HPLC (*p* < 0.001, *r* = 0.994) (Figure 3). Mean rate of sweat Na^+^ loss (54 ± 27 vs. 39 ± 22 mmol·h^−1^) was higher in males than females (both *p* < 0.05) (Figure 2B) but no difference was observed in whole-body sweat [Na^+^] losses between sexes (*p* > 0.05). The majority of runners’ sweat [Na^+^] were classified as moderate (male: 76%; female: 51%). High sweat [Na^+^] was least prevalent as it was only observed in 8% and 16% of male and female runners respectively. Mean sweat [K^+^] did not differ between the male (3.9 ± 1.1 mmol·L^−1^) and female runners (3.5 ± 0.7 mmol·L^−1^) (*p* > 0.05).

## 4. Discussion

The aim of this study was to evaluate the hydration status, fluid intake, sweat rate, and sweat [Na^+^] in recreational tropical native runners. Consistent with previous studies, WBSR ranged between 0.3–2.5 L·h^−1^, with male runners having higher WBSR and WBSL relative to body mass than female runners. Fluid intake was lower in male runners compared to females, who consumed more fluid relative to body mass than males. Predicted whole-body sweat [Na^+^] did not vary between sexes and did not correlate with any other parameters. However, males had a higher sweat Na^+^ loss and rate of sweat Na^+^ loss as compared to female tropical native runners.

The mean WBSR (male: 1.3 ± 0.5 L·h^−1^ and female: 0.9 ± 0.3 L·h^−1^) for tropical native runners was similar to those reported previously in American runners involving 275 male and 52 female adult endurance athletes (male: 1.4 ± 0.4 L·h^−1^ and female: 1.1 ± 0.6 L·h^−1^) (16). Relative WBSL was also comparable to experienced marathon runners in a 16-km race (18.7 ± 7.9 vs. 21.6 ± 5.1 mL·kg^−1^·h^−1^), albeit the race being performed in an environment with cooler ambient temperature but similar relative humidity (29.6 ± 1.2 °C, 70 ± 15% vs. 20.5 ± 0.7 °C, 76.6 ± 1.7%) [38]. These similar observations, despite differences in environmental conditions, could be caused by the effects of heat acclimatisation in experienced athletes [39]. Experienced athletes have thermoregulatory adaptations which include increased sudomotor function associated with acclimatisation [39].

The mean WBSR observed during our study is lower compared to another study involving male runners in a half-marathon, despite similar environmental conditions (1.3 ± 0.5 vs. 1.5 ± 0.3 L·h^−1^) [40]. A reason for this observation is likely to be differences in exercise intensity. The running pace in the present study was slower than that performed in the half-marathon (6.8 vs. 5.6 min·km^−1^). Both laboratory and field-based studies have reported the relationship between exercise intensity and sweat rate in various exercise types such as running, cycling, and football training [41,42]. However, it is important to note that despite the slower running speed, the relative intensity of exercise may have been similar between the two groups. Speculatively, runners completing a half-marathon distance would have a greater maximal aerobic capacity and faster submaximal running velocities compared to tropical native recreational runners. However, the relative intensity of the runs was not measured and hence, was a limitation of the present study.

Our study found that male runners had a higher WBSR than female runners, and that sweat loss relative to body mass was lower in female than male runners. Males generally have a higher sweat rate than females due to larger body size, musculature and higher metabolic rate [43]. The sex of the runner also plays a role in the peripheral control of sweat rate via the number of activated sweat glands (ASGs) and sweat gland output (SGO) [44]. While females have a higher number of ASGs than males during the early follicular phase, their sweat rate in dry and/or humid environments is still lower than males [45]. This may be related to SGO, as previous studies have reported lower SGO in females compared to males during passive heat stress [46]. Moreover, previous evidence on the difference in thermoregulatory adaptation mechanisms revealed that after training, males can only enhance SGO while females can enhance both SGO and ASGs. While trained athletes usually have a higher sweat rate than untrained athletes due to the thermoregulatory adaptations, sweat responses at the same exercise intensity showed that trained males still have higher sweat rates than trained females [44]. Thus, sex can play a role in determining fluid loss through sweat.

To support performance and reduce the risk of heat illness, the American College of Sports Medicine Position Stand advises to avoid body mass loss of >2% from pre-exercise body mass during exercise [12]. In this study, six male and four female runners experienced >2% body mass loss after the run. Consistent with previous research during a marathon race, the mean *ad libitum* fluid intake did not differ between male and female runners [47]. However, when expressed relative to body mass, fluid intake was higher in female runners compared to their male counterparts. As a result, we also observed that female runners experienced less negative net fluid balance relative to body mass. This could be due to the lower pre- and post-exercise body mass of female runners as compared to males. However, there remained a large range (−46–15 mL·kg^−1^·h^−1^) in net fluid balance after exercise in both males and females. This observation would suggest that runners should understand their individual fluid intake requirements to prevent excessive hypohydration, link levels of hypohydration with running performance, and reduce the risk of accumulating body mass through fluid intake during exercise [25]. It is important to note that, due to the nature of the field study, it was not possible to measure body mass in the ideal conditions (i.e., nude body mass) or weigh individual running attires after the run. Based on previous studies, a fully soaked running attire weighs approximately 0.26 kg [48,49]. Although accounting for a trapped sweat volume of 0.26 L or lower would not affect our overall conclusions, this is an acknowledged limitation of the present study.

Predicted whole-body sweat [Na^+^] did not differ between sexes. However, male runners had higher mean rate of Na^+^ loss than female runners. The likely reason for these observations is the higher mean WBSR and WBSL relative to body mass in males than females [50]. This indicates that runners who sweat more are at a greater risk of losing more Na^+^ than runners who sweat less. The predicted whole-body sweat [Na^+^] of tropical native runners ranged between 11–80 mmol·L^−1^. These values are similar to a previous study in runners completing a marathon (7–95 mmol·L^−1^ [36]), albeit at a lower mean ambient temperature (24.4 ± 3.6 °C) and mean relative humidity (28 ± 5%). Our findings are also comparable to the sweat [Na^+^] data collected from 506 athletes across various sports (12.6–104.8 mmol·L^−1^) [16]. Asian populations have higher dietary Na^+^ intake across all ages as compared to those from other parts of the world [51,52]. While the World Health Organisation recommends that sodium intake should not exceed 2 g/day, most Southeast Asian countries consume more than the recommended amount [53]. For example, the average sodium intake in Thailand is 3–5 g per day and the average sodium intake in Singapore is 3–4 g per day [53]. As high Na^+^ intake has been shown to increase sweat [Na^+^] [54], the higher predicted whole-body sweat [Na^+^] in the present study may have been expected. The analysis of sweat [K^+^] in the present provided a quality control for the analysis of sweat samples [15]. Therefore, we have confidence in the sweat [Na^+^] values. A possible explanation for the similarity in sweat [Na^+^] might be the effects of heat acclimatisation, which has been shown to reduce sweat sodium [55,56]. However, dietary intake and its associated Na^+^ intake were not measured, and this is a limitation of the present study. Thus, it is not possible to ascertain if acclimatisation counteracted the impact of high Na^+^ intake on sweat [Na^+^]. Given the inter-individual variability of predicted whole-body sweat [Na^+^] and that no correlation to age, exercise intensity, exercise duration or sweat rate was found, the results support recommendation of individualised fluid and electrolyte replacement strategies.

Runners were more likely to be hypohydrated before the morning run compared to the evening. It is recommended that participants should slowly drink approximately 5–10 mL per kg of their body mass at least 2 h before exercising to allow for fluid absorption and achieve euhydration [22]. The finding in the present study could be attributed to insufficient time to drink appropriate volumes of fluid and the ingestion of food in the morning. The moderate positive correlation between pre-exercise USG and fluid intake, and moderate negative correlation between pre-exercise USG and WBSL suggest that hypohydration may impair thermoregulatory function, which increases the risk of heat illness in hot climates [57]. This is of interest as endurance running events such as marathons typically begin early in the morning. Since achieving euhydration before exercise can reduce the risk of heat illnesses [9,12], runners are encouraged to adjust their pre-race wake-up time to ensure they can meet pre-exercise hydration guidelines when running in tropical climates.

### Practical Implications and Future Directions

This study showed that the range of sweat rates and sweat [Na^+^] of tropical native runners were similar to that reported previously in hydration literatures. Therefore, individual fluid recommendations also apply to tropical native runners. Future studies should aim to analyse the runners’ dietary Na^+^ intake to understand the impact on the sweat Na^+^ losses. Finally, the data in the present study were collected during recreational running. Understanding the impact of hydration strategies before and during exercise on running performance in competitive tropical native runners is required. Correspondingly, these studies can ascertain if the threshold of hypohydration tolerance, i.e., 2% body mass loss on exercise performance, also applies to tropical native athletes.

## 5. Conclusions

These descriptive data gathered on recreational tropical native runners revealed the individual variability in hydration status, fluid intake, sweat rate, and sweat Na^+^ loss during exercise. Female runners experienced less negative net fluid balance compared to males due to greater fluid intake per body mass and lower sweat loss. Whole-body sweat [Na^+^] also varied between individuals independent of sex.

## Figures and Tables

**Figure 1 nutrients-13-01374-f001:**
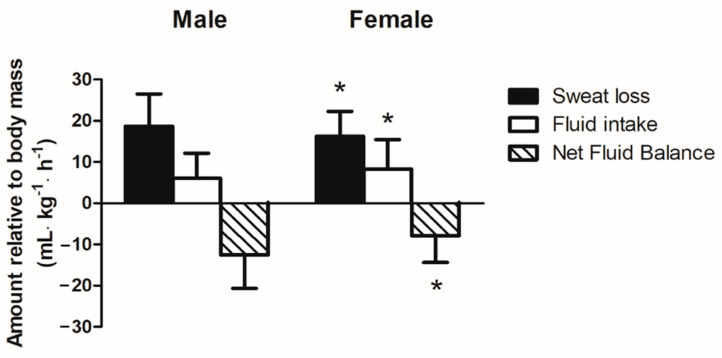
Relative sweat loss, fluid intake, and net fluid balance (mL·kg^−1^·h^−1^) between sexes. Data are presented as mean ± SD. * *p* < 0.05, compared to male participants.

**Figure 2 nutrients-13-01374-f002:**
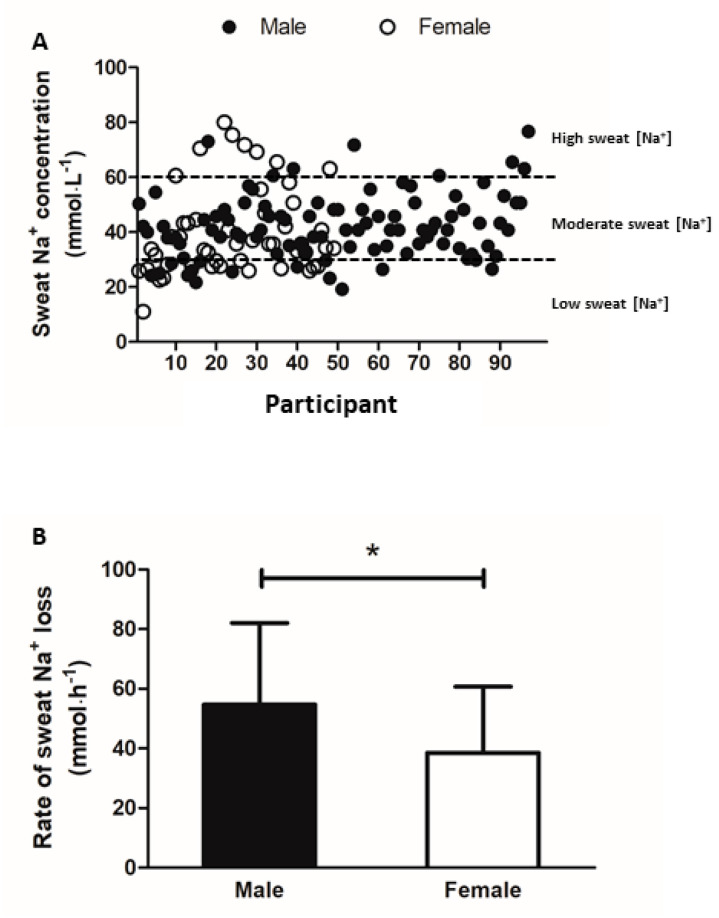
(**A**) Distribution of whole-body sweat Na^+^ concentration (mmol·L^−1^) among 97 male and 49 female participants; (**B**) rate of sweat Na^+^ loss (mmol·h^−1^) compared between male and female runners. Data are presented as mean ± SD. * *p* < 0.05, compared to male participants.

**Figure 3 nutrients-13-01374-f003:**
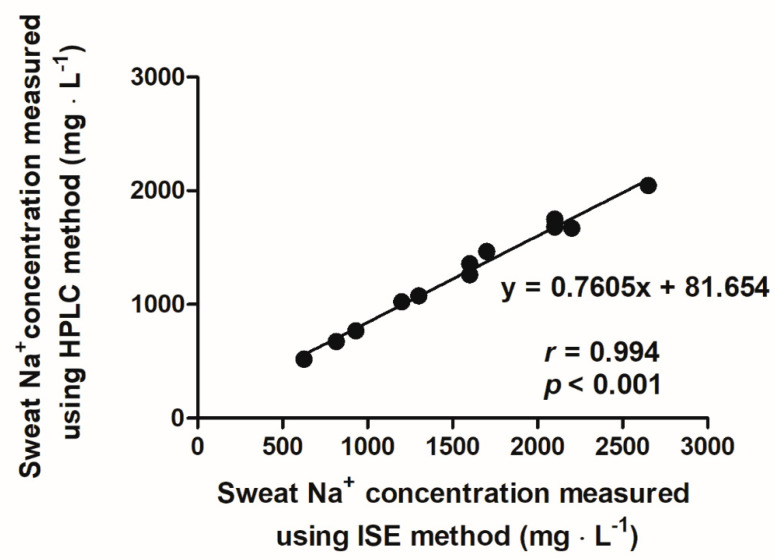
Relationship between sweat Na^+^ concentration measured by ion-selective electrode (ISE) and high-performance liquid chromatography (HPLC) methods.

**Table 1 nutrients-13-01374-t001:** Time of day, time, environmental conditions and number of participants during each running session.

Session	Time of Day	Time	Mean Ambient Temperature (°C)	Mean Relative Humidity (%)	Participants	
					Male	Female
1	Morning	7.00 a.m. to 8.00 a.m.	28.5	75	9	6
2	Morning	6.00 a.m. to 8.00 a.m.	30	86	10	4
3	Evening	7.00 p.m. to 8.00 p.m.	29.5	63	16	8
4	Evening	7.00 p.m. to 8.00 p.m.	28	87	31	13
5	Evening	7.00 p.m. to 8.00 p.m.	29.8	56	12	12
6	Evening	5.00 p.m. to 6.00 p.m.	31.5	55	24	21

**Table 2 nutrients-13-01374-t002:** Mean age, running profile, body mass change, sweat rate, and fluid intake of male and female runners across all running sessions. Data are presented as mean ± SD (range).

	Male (*n* = 102)	Female (*n* = 64)
Age (years)	36 ± 9 (21–68)	34 ± 9 (22–62)
Running duration (min)	43.7 ± 14.8 (33–97)	43.6 ± 13.6 (43–100)
Running distance (km)	6.4 ± 1.1 (2.5–12.5)	5.3 ± 1.1 * (2.5–10)
Running pace (min·km^−1^)	6.8 ± 3.7 (3.5–10.0)	8.2 ± 3.9 * (4.3–11.0)
Pre-running body mass (kg)	70.8 ± 10.6 (46.9–99.7)	56.7 ± 9.7 * (42.3–93.2)
Post-running body mass (kg)	70.2 ± 10.6 (46.6–99.0)	56.4 ± 9.6 * (42.3–92.8)
Percentage body mass loss (%)	1.3 ± 0.5 (0.2–3.6)	1.2 ± 0.5 (0.1–3.7)
Whole-body sweat loss (WBSL) (L)	0.9 ± 0.3 (0.2–2.6)	0.6 ± 0.3 * (0.1–1.9)
Whole-body sweat rate (WBSR) (L·h^−1^)	1.3 ± 0.5 (0.2–3.8)	0.9 ± 0.3 * (0.1–2.2)
Fluid intake (L)	0.3 ± 0.3 (0–1.1)	0.3 ± 0.2 (0–1.1)

* *p* < 0.05, compared to male participants.

**Table 3 nutrients-13-01374-t003:** Level of dehydration between sexes (male vs. female) and time of day of running session (morning vs. evening) based on pre-exercise urine specific gravity (USG).

Time of Day of Session	Sex	Urine Specific Gravity (USG)		
		≤1.020	>1.020	>1.030
Morning	Male	11 (69%)	4 (31%)	1 (6%)
	Female	7 (78%)	2 (22%)	0 (0%)
	Total	18 (72%)	6 (28%)	1 (4%)
Evening	Male	59 (85%)	10 (15%)	1 (1%)
	Female	43 (84%)	8 (16%)	0 (0%)
	Total	102 (85%)	18 (15%)	1 (1%)

## Data Availability

The data presented in this study are available on request from the corresponding author and the permission of all parties involved in the study.

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
