# Peer review of "Hydration Status, Fluid Intake, Sweat Rate, and Sweat Sodium Concentration in Recreational Tropical Native Runners"

_nutrients, 2021, doi:10.3390/nu13041374_

Round 1

Reviewer 1 Report

The paper "Hydration status, fluid intake, sweat rate, and sweat sodium concentration in recreationally trained tropical native runners" aimed to investigate the sweat rate and the fluid balance during a training session in tropical native runners. The authors started from the point that this population might be acclimatised to live in such a hot and humid environment (tropical climate), so they wanted to observe and then compare the sweat rates of these recreational athletes and to compare the results with those of other populations in the literature, and the actual consensus recommendations.

The paper is globally well written, and both the introduction and discussion sections seem appropriate. Although the topic might sound interesting, I have some points that need to be discussed.

  • Methods, Sect 2.2: What do you mean by "Participants ran at their individual pace"?. It was an "easy" pace, a moderate-intensity pace, a race-pace? Saying that pace was "individual" does not say anything regarding the intensity of the run, that as you state in the discussion strongly influence sweat production.
  • Methods, Sect 2.2: "A water station was provided. Runners consumed water ad libitum from individual water bottles." Do the runners usually bring with them a water bottle during training sessions? In my experience, most of them do not. Should be this point discussed? I mean, if the aim is to investigate a "real context" situation, with people running at a self-selected pace as they would do on a daily basis, and they normally do not have a water station available during their training sessions, this could be a problem for the practical applications of the results of the study. 
  • Methods, Sect 2.2, Lines 109-110: "participants wore the same attire as during the pre-exercise...". Do they changed or removed the wet t-shirt in the post-exercise weight?
  • Results, Sect 3.2: "running sessions were between 30-100min" However, in the abstract, you reported 45-70, and in the methods 30-70 (excluding warm-up and cool down). Please, make this point clear and coherent in the manuscript.
  • Results, Table 2: I would like to know more characteristics of the participants, more than just age: height, BMI, number of training per week, or weekly/monthly/yearly training volume. This is important to define the word "recreational runners". Looking at the running pace, it seems that 6.8 min/km for males and 8.2 min/km for females are really slow paces, even for recreational runners! So, I think this point is questionable because, as I pointed out before, the intensity affects the sweat rate, and then the results of the paper. Does this speed include walking phases during warm-up and cold down, maybe? Or did participants run at very very low intensity? Or their fitness level was bad, so the term "recreational runner" is inappropriate? Please, discuss this point. 
  • Discussion, Lines 215-216: Need a reference here. 
  • Discussion, Lines 261-263: "In this study, we found six male and four female runners experienced > 2% body mass loss during recreational running. This was a consequence of the water availability and opportunity to drink during the runs." What do you mean? As reported here, this sentence makes no sense to me. Maybe it the opposite, that subjects did not experience >2% BM loss precisely because of the water availability during the run, a thing that is probably not so common while they usually train. 

In light of all the above-reported comments, I suggest a major review for this paper before considering it for a possible publication. 

Reviewer 2 Report

The manuscript is definitely set up well. However, there are some critical issues that should be addressed.

In this manuscript, the authors attempted to investigate the effect of recreational running of the natives in high temperature and humidity on the rate of sweating and sweat sodium loss. The study is valuable because many people want to participate in races (marathons, half-marathons) organized in countries with tropical climates. Therefore, it is important to know how to properly prepare the participants for this type of effort due to the correct hydration of their body.

However, as it stands, several shortcomings prevent the publication of this manuscript in the journal. My detailed comments are as follows:

ABSTRACT

Page1-line 22: What do you mean "Demographic data"?

Page1-line 22: "running profiles": Use more adequate phrase. What exactly you want to describe using this expression.

Page1-line 24: "A p-value cut-off of 0.05 depicted statistical significance." This sentence is not necessary in the summary. This is a rather obvious assumption of the research and you just need to explain it in the "Statistical analysis" section.

Page1-lines 26-27: "Males had higher whole-body sweat rate (1.3 ± 0.5 L·h-1 vs 0.9 ± 0.3 L·h-1, p < 0.05) than females." This statement is only a confirmation of a well-known fact.

Page1-lines 28-29: "Sweat profile and composition in tropical native runners are similar to reported values in the literature." What is the novelty look in the present manuscript?

INTRODUCTION

Page1-line 36: "(1, 2)": Which part of these manuscripts presents results described by you: "ten- to twenty-fold"

Page1-line 38: "(3-6)": Check carefully how you write your references. Apply the journal's guidelines, e.g.
J Sports Sci.;
Medicine and Science in Sports and Exercise.;  
Journal of applied physiology.

Page1-lines 40-41: "electrolyte imbalance": At the beginning of the "Introduction" you should shortly describe more detail about electrolyte imbalance. Not only about sodium ions but also others (potassium, chloride, zinc).

MATERIALS AND METHODS

Page2-line 77: What does is mean "an ecological design"?

Page2-lines 83-84: 1. Did you measured resting heart rate after running?; 2. Using ear thermometer you cannot directly measure core temperature. Familiarize yourself with the methods of measuring core body temperature. In my opinion, it is proper to define the measurement as body temperature.; 3. Were the measurements done directly after running?

Page2-lines 94-95: "public park with a 2.5 km running track": Describe the place. Region and country.

Page2-line 95: "consumed water ad libitum": What kind of water the runners consumed? What concentration of sodium and potassium ions were within?

Page2-line 96: "individual water bottles": Did the runners drink the same type of water?

Page3-lines 110-111: "Post-exercise body mass was recorded to the nearest 0.10 kg": Repeated information (lines 101-102).

Page3-lines 113-114: VERY IMPORTANT! Why authors don't present potassium concentration if these ions were measured?

Page3-line 119: Urine specific gravity (USG) – add abbreviation

Page3-lines 122-123: Analyse this fragment text carefully. Use other description for example USG > 1.020 indicating "moderate" hypohydration?

Page3-line 128: Describe the model of used analyser.

Page3-lines 132-133: Why only 12 samples were analysed? How many analysed samples belong to low, moderate and high group?

Page3-lines 133-134: Describe the type of equipment HPLC used to determination of ions concentration.

Page4-line 139: If you analysed differences between males and females you shouldn`t use t-test. The groups were of different sizes. What test was used to evaluate the determination of the normal distribution (Shapiro-Wilk test)? If you used t-test the distribution of group should be normal.

RESULTS

Page4-line 150: All six sessions were realized in the same day? Unfortunately it is not clearly described.

Page4-line 154: Did the authors used any subjective fatigue scale, such as the Borg Scale (RPE)?

Page4-line 157: "between 30–100 min": If the duration of each session is exactly 60 minutes (for example, 7:00 p.m. to 8:00 p.m.), how do some participants reach 100 minutes?

Page4-line 163: "(Table 2)": Explain the abbreviations (WBSL, WBSR) below the table. Tables must be self-describing.

Page5-line 167: Unfortunately, the number of urine samples does not match. Adding the lack of samples and analyzed (Table 3) it do not obtain the sum 166.

Page5-lines 186-188: "With regards to fluid intake during running, 15 of the 102 male runners (14.7%) did not drink any water but only two of the 64 female runners (3.1%) did not drink any water during running.": Rebuild this sentence.

Page5-line 189: "(Table 1).: Check carefully again. Are you sure it is Table 1?

Page5-line 191: "(6 ± 6 vs 8 ± 7 mL·kg−1·h−1, p < 0.05)": The data is inconsistent with Figure 1. The standard deviation differs significantly in both places.

Page6-line 203: In my opinion it is worth to demonstrate the figure presenting this correlation. Think about it. This is a valuable methodological result of this research. It follows that both methods are very similar and can be used interchangeably.

DISCUSSION

Page7-lines 262-263: "This was a consequence of the water availability and opportunity to drink during the runs": This sentence is not fully understood. You wrote that the participants had unlimited access to water. Did (exactly) these participants you describe declared not consuming fluids during the run?

Page7-lines 283-284: "Asian populations have higher dietary Na+ intake across all ages as compared to those from other parts of the world (49, 50). ": Present the average consumption of sodium by Asian population in comparison to other populations.

Round 2

Reviewer 1 Report

Dear Authors, 

I recognize your effort in responding to all my previous comments.

However, I still have concerns regarding:
a) the fitness level/running experience of the participants, which in my opinion do not represent a cohort of "trained runners", in terms of no. of training per week (1-2 sessions), confirmed from the really slow training speed, even if at low intensity.
b) the fact that this private group usually has a water station available at their training sessions is uncommon. I believe that most recreational runners worldwide do not have this "service" during their training. This point is maybe critical for the application of the results to all the recreational runners population.
c) Although the participants dried themselves, I think that keeping the wet (and heavy) t-shirt in the post-weight could have influenced your results. In particular, considering the % reduction of body mass, I could imagine it could have been substantially higher than the one you actually measured, so this could be a key point for the discussions.

Reviewer 2 Report

I thank the authors for their efforts to correct the manuscript. Nevertheless, there are still a few things to improve.

Title. In my opinion, in the title of the manuscript, you should include information regarding the determination of potassium in sweat. “Hydration status, fluid intake, sweat rate, and sweat sodium and potassium concentration in recreationally trained tropical native runners”. The title will be more appropriate to the research carried out and the results contained in the manuscript.

I still feel that the use of the expression "Demographic data" is not appropriate. It has a different meaning and applies to the entire population or ethnic group. You studied a selected group of people with a similar lifestyle. You have to use other terms. Please consider whether the use of this expression is necessary to describe the studied group of people.

L41. "ten- to twenty-fold". Please present excerpts from the publications [1.2] mentioned by you in which such data are provided. Perhaps I missed something.  

L44-45. In my opinion, you have approached the subject of sweat composition too broadly. Since you are involved in the analysis of sweat during exercise, you should briefly describe its composition. The consequence of an increase in the rate of sweating is not only hyponatremia and hypokalemia.

I still feel that the use of the expression "an ecological design" is not appropriate. You are using a broad term referring to a limited research group.

Please read:

https://www.nature.com/articles/6400454.pdf?origin=ppub

L127. Please specify the research equipment used. You describe that you are measuring potassium in sweat and you are only mentioning a sodium analyzer. Be more precise and specific when describing the methods.

“Only 12 were analysed because we used a sample of the total number available (12 in this case) for our comparison against the gold standard (HPLC).”

I find your explanation unconvincing. I don't understand why you chose 12 samples and what was the rationale behind the exact samples. Was it a random selection or did you have any reasons for selecting them?

L166 and Table 3. In the " Study design and participants" section, 166 participants (“A total of 102 males and 64 females”) are indicated. Unfortunately, the number of participants is still incorrect (n = 168). I am beginning to suspect that the data may be manipulated.

Figure 1. Thank you very much to the authors of the manuscript for considering my suggestion. However, I must point out that there must be an identical scale on both axes. In addition, you should put an identity line or possibly a curve equation.
